# Mixture Design and Mechanical Properties of Recycled Mortar and Fully Recycled Aggregate Concrete Incorporated with Fly Ash

**DOI:** 10.3390/ma15228143

**Published:** 2022-11-17

**Authors:** Lijuan Zhang, Dong Ding, Jun Zhao, Guosen Zhou, Zhi Wang

**Affiliations:** 1School of Mechanics and Safety Engineering, Zhengzhou University, No. 100 Science Avenue, Zhengzhou 450001, China; 2Forth Co., Ltd. of China Construction Fifth Division, 69 Shihua Road, Guancheng District, Zhengzhou 450004, China

**Keywords:** recycled mortar, recycled concrete, mixture proportion method, mechanical properties, fly ash

## Abstract

Recycled aggregate concrete (RAC) is a sort of green, low carbon, environmental protection building material, its application is of great significance to the low carbonization of the construction industry. The performance and strength of RAC are much lower than natural aggregate concrete (NAC), which are the key factors restricting its application. Class F fly ash is a cementitious material that is considered environmentally hazardous. In this paper, appropriate water-binder (*w*/*b*) ratios were found through a mortar expansion test at first. The compressive strength of recycled mortar incorporated with class F fly ash was further studied. On this basis, the mechanical properties of nine groups of fully recycled aggregate concrete (FRAC) with a *w*/*b* ratio of 0.3, 0.35, and 0.4, and fly ash replacement ratios of 0, 20%, and 40%, were studied. The influence of the *w*/*b* ratio and fly ash replacement ratio on mechanical properties was analyzed and compared with previous research results. In addition, the conversion formulas between the splitting tensile strength, flexural strength, and compressive strength of FRAC were fitted and established. The research results have a certain guiding significance for the mixture design of FRAC and further application of class F fly ash.

## 1. Introduction

Concrete is essential in building construction and infrastructure, and the global demand for concrete is constantly on the rise. The main constituent materials of concrete, river sand and gravel, are non-renewable resources and are extracted most, even more than fossil fuels [1]. Approximately 30–50 billion tons of sand are consumed annually worldwide, and the current rate of extraction of river sand and natural aggregates far exceeds the rate of natural renewal, but cannot fully meet the demands of engineering construction [2]. Meanwhile, a massive amount of construction and demolition (C&D) waste is produced as concrete structures are demolished and disposed of in landfills [3]. An important way to recycle concrete materials is fabricating C&D waste as recycled aggregate (RA) to manufacture recycled mortar and recycled aggregate concrete (RAC). The application of RA can not only consume C&D waste and solve the urban environmental issue contributing to C&D waste, but also reduce the mining and use of natural sand and gravel, protect the natural environment, and solve the current natural resources dilemma. However, the performance of RA changes dramatically due to complex sources and diverse reconstruction techniques of C&D waste.

Currently, concrete materials that use exclusively recycled coarse aggregate (RCA) and recycled fine aggregate (RFA) are referred to as fully recycled aggregate concrete (FRAC). Generally speaking, the surface of RCA is wrapped with a layer of mortar, with a high content of needles and flakes, while the crushing index, the water absorption rate, and the void ratios are larger than for natural coarse aggregate (NCA), and the apparent density is smaller [4]. The chemical composition of RCA and NCA is also different. The content of SiO_2_ in RCA can hit 50% to 60%, while the chemical composition of calcium (CaO) in RCA is mostly below 20%. The NCA is mostly composed of CaO (49.16%) and SiO_2_ (5.46%) [5,6]. RCA, as a siliceous aggregate, shows clearly inadequate water resistance [7]. Thus, recycled aggregate concrete (RAC) generally performs worse than natural aggregate concrete (NAC), with workability [8], density, and mechanical properties [9] decreasing as the replacement ratio of RCA increases. RFA is determined as having a higher water absorption and a more adherent mortar in surface compared to natural fine aggregate (NFA), resulting in in the application of RFA being greatly limit [10]. RFA is firmly excluded from the production of concrete and mortar in most countries and territories [11]. The chemical composition of RFA is similar to that of RCA, with SiO_2_ content being between 60% and 80%, and CaO content being between 5% and 20%. Besides, RFA also contains a certain amount of Al_2_O_3_ (below 10%) and Fe_2_O_3_ (below 10%) [12]. Previous studies in our laboratory have shown that the main mineral compositions of RFA are quartz (SiO_2_), calcite (CaCO_3_), albite (Na_2_O·Al_2_O_3_·6SiO_2_), and anorthite (CaO·Al_2_O_3_·2SiO_2_). CaCO_3_ in calcite can react with C_3_A in cement to form calcium calcium aluminate (C_3_A·CaCO_3_·11H_2_O), thus hindering the formation of calcium hydroxide (Ca(OH)_2_) and hydrated calcium sulphoaluminate (Aft), which improves the compactness of the interfacial transition zone (ITZ) and enhances the bonding property of the interface between the aggregate and cement paste, thus increases the strength of the mortar [13]. Besides, the rough irregular surface of RFA effectively enhances the mechanical interlocking of ITZ between RFA and the cement matrix, which improves the splitting tensile strength [14]. The Ca(OH)_2_ groups in RFA are found to react with fly ash to form secondary hydration compounds that increase the mechanical strength of the mortar [12,15]. Based on the above results, RFA can be more fully applied by an appropriate mixture design method. At present, the main mixture design method of RCA is the additional water consumption method. However, the study in our laboratory showed that, when RFA is used, the mechanical properties of RAC decrease with an increasing RFA replacement ratio and additional water consumption. When the RFA replacement ratio is 100%, the elastic modulus is reduced by up to 20% [14]. Therefore, a new mixture design method should be proposed to obtain FRAC with better mechanical properties. The mechanical properties of concrete are closely related to its workability, especially slump. For FRAC, the slump of fresh FRAC is more difficult to control due to the high water absorption of RCA and RFA. The fluidity of mortar directly affects the slump of concrete, which is mainly affected by the water-binder ratio and sand-cement ratio. Therefore, the fluidity of recycled mortar can be used as a main reference in the mixture design of FRAC.

The water-binder (*w*/*b*) ratio is an important parameter affecting the workability and mechanical properties of concrete. For RAC, RA will absorb a large amount of water so that the effective water in the mixing process will be significantly reduced; using the unilateral water consumption of NAC to calculate the mix proportion of RAC will lead to a significant decrease of RAC slump [16]. Therefore, additional water and admixtures are needed to optimize the performance of the RAC. Wagih et al. [17] reported that a 13% increase in water consumption is required for RAC to achieve a similar performance as NAC. Wang et al. [18] suggested that, when calculating the mix proportion of RAC, in addition to the water consumption designed in accordance with the NAC ratio, he considered the additional water consumption required to make RA reach the saturation state of water absorption, called the additional water consumption method. Before pouring RA, soak RA to make it reach the saturation state of water absorption [19]. However, in the actual pouring process, both methods are not very feasible. For FRAC, since the fine aggregate used is RFA, its unilateral water consumption is even higher than plain RAC, thus the *w*/*b* ratio of FRAC deserves more attention.

Fly ash, one of the main waste products of coal-fired power plants, is considered as being deleterious material due to the significant variability of its leaching characteristics regarding heavy metals [20]. However, fly ash has a good adsorption capacity for NO_2_, SO_2_, organic compounds, and mercury [21], so it is usually used as a supplementary cementitious material in the construction industry [22]; itis concluded that it can improve the workability, mechanical properties, microstructure, and durability of concrete, and meanwhile reduce the heat of hydration and the cracking tendency [23]. In addition, adding liberal fly ash at a low *w*/*b* ratio is an effective method for fabricating high performance concrete [24,25]. The hydration of fly ash and Ca(OH)_2_ groups in RFA ameliorates the adverse effect of RFA on mechanical properties [12,15]. The effect of fly ash on concrete is greatly affected by its characteristics, especially the content of calcium oxide and fineness. The American Society for Testing Materials (ASTM) specification C618-19 [26] defines two classes of fly ash (classes C and F). As the total sum of SiO_2_, Al_2_O_3_, Fe_2_O_3_, and SO_3_ exceeds 70% of the total volume, it is considered as class F, otherwise it is class C. The Chinese standard GB/T 1596-2017 [27] is more intuitive for defining the class of fly ash; fly ash is defined as class F if the CaO content is less than 10%, otherwise it is class C. China is the world’s largest producer of fly ash, with an annual output of 600 million tons, but only 70% of fly ash is fully utilized. Moreover, the CaO content of most fly ash produced in China is between 1.1% and 7% [28], belonging to class F fly ash. Thus, the application of class F fly ash should be further studied.

Sand ratio also influences the mechanical properties and workability of concrete. In the design of an ordinary concrete ratio, it is considered that the essence of determining the best sand ratio is to determine the best sand-cement ratio. As the surface of RCA generally adheres to a layer of old mortar, if the sand ratio of RAC is calculated according to the calculation method of NAC, it will make its content of mortar much higher than NAC, which is one of the major reasons for the deterioration of the performance of RAC. In addition, the particle morphology of RFA and NFA is also very different; if the outer surface of RFA is rough, it will further affect the workability of RAC.

At present, the strength of FRAC can be improved by reducing the water-binder ratio, however, there is no effective method to control the workability of FRAC. The workability not only affects the application of FRAC in practical engineering, but also plays a decisive role in its strength, compactness, and durability after hardening. Therefore, how to determine the mix design parameters of FRAC to ensure the workability and strength of RC is the focus of this paper. A new mixture design method of FRAC is provided. The workability of FRAC can be significantly improved by an optimal sand-cement ratio and sand ratio, and the effects of the water-binder ratio and fly ash replacement ratio on FRAC made by this method were studied, which provides the experimental basis and theoretical guidance for the mixture design of FRAC and the application of class F fly ash.

## 2. Materials and Methods

### 2.1. Mixture Design Method

In order to ensure the workability of FRAC, a new mixture design method was conducted based on the fluidity of recycled mortar. The determination process of the main parameter mixture design is as follows:

Firstly, the optimal sand-cement ratio corresponding to different water-binder ratios was obtained through the expansion test of recycled mortar. Secondly, the sand ratio was calculated through the “Mortar abundant coefficient method”. This method considers that, in addition to filling the void between the CA, FA also needs to have a certain amount of surplus to form the cement mortar and ensure that the fresh concrete mix has enough flow performance. Finally, the water content, cementitious materials, and RCA and RFA of FRAC under different water-binder ratios can be directly obtained by the “Absolute volume method”.

On this basis, the influence of different water-binder ratios and fly ash replacement ratios on recycled mortar and FRAC made by this method was studied. The main research steps of the paper are shown in Figure 1. Based on this mixture design method, not only FRAC, but also recycled mortar with good workability can be prepared.

### 2.2. Materials

Grade 42.5 ordinary Portland cement and fly ash were used as binder materials. The physical properties and main chemical composition of cement and fly ash are listed in Table 1 and Table 2, respectively. The particle size distribution of both binder materials are listed in Table 3 and Figure 2. The particle size of fly ash is larger than that of cement. The CaO content of fly ash is 3.36% and the total sum of SiO_2_, Al_2_O_3_, Fe_2_O_3_, and SO_3_ is 91.303%. Therefore, the fly ash used in the paper belongs to class F, according to C618-19 [26] and GB/T 1596-2017 [27]. The fly ash replacement ratio, which is the ratio of the weight of fly ash to the weight of total cementitious material, is taken as 0, 20%, and 40%. RCA and RFA are derived from discarded laboratory concrete specimens. The waste concrete was crushed and screed to prepare as RCA (5–20 mm) and RFA (0–4.75 mm). The material properties of RCA and RFA are tested by the Chinese standard JGJ 52-2006 [29] and shown in Table 4. The crush index of RCA is 14.1%, which is higher than the crush index of NCA. The fineness modulus of RFA is 3.15, which is larger than 3, and therefore belongs to coarse sand. The surface of RFA is rougher than the NFA. The water-reducing rate of polycarboxylic acid water-reducing agent used in the experiment is 27%.

### 2.3. Specimen Preparation

Owing to the large water absorption of RCA and RFA, the workability of FRAC will be seriously affected if made using the mix design of ordinary concrete. In order to ensure that the slump of the FRAC can reach more than 50 mm, the expansion test of the recycled mortar should be carried out first. The expansion test was carried out by the Chinese standard GB 50119-2013 [30]. The detailed method is to pour the mortar into the mortar expansion cylinder twice (each time the amount is 1/2 of the cylinder height) and use the pounding rod along the edge to the center clockwise to evenly pound 15 times. After pounding and scraping, the cylinder is slowly lifted and the maximum diameter in two directions perpendicular to each other is measured with a steel ruler after 10 s, and the average value is taken as the mortar expansion. The *w*/*b* ratios of mortar were selected as 0.3, 0.35, 0.4, 0.45, and 0.5 and adjusted the sand-ash ratio to make the expansion of the mortar reach more than 300 mm. In addition, on this basis, the fly ash replacement ratio was 0, 20%, and 40%.

The detailed mix design of FRAC needs to be determined by the properties of the recycled mortar. The mixing process of FRAC is as follows: firstly, the cement and RFA were mixed without water for 120 s. In the second step, the water-reducing agent and half of the water were added and mixed for two 120 s so that the cement and water-reducing agent could fully react. Point three, the RCA and the remaining half of water were added and mixed for 120 s. The slump was tested first, then the concrete was loaded into the molds and vibrated on a vibrating table for 90 s, cured indoors for 24 h, then demolded and placed in a standard curing room. After 28 days of curing, the mechanical properties of FRAC were tested.

### 2.4. Test Methods

Standard mortar test blocks with a side length of 70.7 mm were used to test the compressive strength of recycled mortar, three test pieces were in each group and the loading speed was 0.5 KN/s. The compressive strength and splitting tensile strength of FRAC were tested with 100 mm^3^ test blocks, with three test pieces in each group, and were performed by a digital display pressure tester. The loading speed of the compressive strength test was 0.5–0.8 MPa/s, and the loading speed of the splitting tensile strength test was 0.08 MPa/s. As the size of the specimens used is smaller than the standard specimen block with a side length of 150 mm, the compressive strength and splitting tensile strength are multiplied by the discount factor of 0.95 and 0.85, respectively. Beams with a side length of 100 mm × 100 mm × 400 mm were tested for flexural strength with a loading speed of 0.1 mm/min. Mortar specimens and FRAC specimens were maintained in a standard maintenance room for 28 d before testing. The mechanical properties testing equipment and methods of mortar specimens and FRAC specimens meet the requirements of the Chinese standard JGJ/T70-2009 [31] and GB/T50081-2019 [32], respectively.

## 3. Properties of Recycled Mortar

### 3.1. Effect of Sand-Cement Ratio

According to Table 5, the variation relationship between the expansion of mortar and sand-cement ratio under various *w*/*b* ratios is shown in Figure 3. Figure 3 shows that the mortar’s sand-cement ratio should decrease as the *w*/*b* ratio decreases.

The expansion of the highest sand-cement ratio for each group of tests is only 150–170 mm, which is attributed to the high water absorption of RFA. Therefore, a large amount of water will be absorbed so that the actual amount of water involved in the hydration reaction of the cement is much smaller, resulting in a greatly reduced amount of cement paste, and the water-reducing agent also does not fully react with cement. With the reduction of the sand-cement ratio, it is obvious to observe the air bubbles generated by the reaction of the water-reducing agent and cement when mixing the mortar, which shows that in the production of recycled mortar, it is important to ensure an effective amount of water and a sufficient mixing time in order to make the water-reducing agent take effect and ensure the fluidity of mortar.

The expansion of mortar increases with the decrease of the sand-cement ratio, with slow changes in the early stage. As the sand-cement ratio continues to become small, the curves of all five groups will have an inflection point, after which the expansion will suddenly become larger. Large expansion will reduce the strength of both mortar and concrete. Meanwhile, the sand-cement ratio decreases and the content of cement in the mortar increases, which will increase the economic cost. In an effort to ensure the strength and workability of FRAC, the sand-cement ratio should be maximized after satisfying the mortar fluidity, while ensuring optimal economic efficiency. Therefore, the maximum sand-cement ratio of expansion over 300 mm is selected as the test optimal sand-cement ratio.

In this test, three *w*/*b* ratios of 0.3, 0.35, and 0.4 were selected, and the optimal sand-cement ratios of 1, 1.1, and 1.3 were selected under the three *w*/*b* ratios. The fly ash replacement ratio was taken as 0, 20%, and 40%. Subsequently, an expansion test and a compressive strength test of the recycled mortar were executed. The detailed mixture design and results are shown in Table 6. From Table 6, the compressive strength of plain recycled mortar under three water-binder ratios is 92.0%, 85.7%, and 90.0% of standard mortar produced with high quality standard sand, respectively, which proves that using this mixture design method can produce recycled mortar with high strength.

### 3.2. Effect of Fly Ash Replacement Ratio

The variation of expansion with the fly ash replacement ratio is shown in Figure 4. As the fly ash replacement ratio increases, the expansion of the recycled mortar increases continuously, which shows that fly ash can effectively increase the fluidity of recycled mortar. This is because the shape of fly ash is mostly spherical [33], and the particle size of fly ash is also larger than cement. The addition of fly ash helps to improve the particle size distribution and the fluidity of mortar. In addition, fly ash reduces the flocculation of cement particles and plays a role in dilution [34]. However, when the *w*/*b* ratio is high, the increase of expansion is decreasing with the increase of the fly ash replacement ratio, which shows that the improvement effect of fly ash on the workability of mortar is more obvious when the *w*/*b* ratio is small. When the *w*/*b* ratio is 0.3, the increase of expansion from 300 mm to 400 mm is the most obvious as the fly ash replacement ratio increases from 0 to 40%.

The variation of the compressive strength of recycled mortar with the fly ash replacement ratio under different *w*/*b* ratios is shown in Figure 5. The strength of recycled mortar decreases continuously with the increase of the fly ash replacement ratio. Since the active substances in fly ash that can participate in the hydration reaction are much smaller than those in cement, the increase of fly ash admixture will reduce the degree of the hydration reaction of the whole cementitious material, thus reducing the compressive strength of mortar. Moreover, with the increase of the *w*/*b* ratio, the decreasing effect of fly ash on compressive strength is more obvious. When the *w*/*b* ratio is 0.3, the compressive strength only decreases by 1.9%. When the *w*/*b* ratio increases to 0.4, the compressive strength is reduced by 34.1%. It can be seen from Figure 6 that the compressive strength of recycled mortar generally shows a downward trend as the *w*/*b* ratio increases, and the decreasing amplitude of compressive strength increases with the increase of the fly ash replacement ratio.

In summary, although fly ash can ameliorate the fluidity of mortar, it will reduce the compressive strength. Especially for mortar with a larger *w*/*b* ratio, the effect of fly ash to ameliorate the fluidity of mortar is reduced, and the reduction of compressive strength is increased. Thus, for mortar with a high *w*/*b* ratio, it is not suitable to mix a large magnitude of fly ash. In the meanwhile, as the fly ash replacement ratio is high, the compressive strength of recycled mortar also decreases more quickly as the *w*/*b* ratio increases.

## 4. Mix Design and Properties of FRAC

### 4.1. Determination of Sand Ratio

Generally speaking, the fine aggregate needs to fill the void between the coarse aggregate first, and then form mortar to wrap the surface of the coarse aggregate, so the sand ratio of FRAC can be calculated by the following two formulas.
(1)$$V_{s}=\gamma \times V_{g}\times P_{g}$$
(2)$$\beta_{s}=\frac{m_{s}}{m_{s}+m_{g}}$$
where, *V_g_*, *V_s_* are the volumes of the coarse aggregate and fine aggregate per cubic concrete, respectively. *P_g_* is the void ratio of the RCA, which is 52.8% in this test. γ is the mortar rich coefficient, which is the ratio of the volume of mortar to the void volume of coarse aggregate; γ = 1.1–1.4 for ordinary concrete [28]. In the test, the value of γ is taken as 1.3.

*β_s_* is the sand ratio. *m_g_*, *m_s_* are the mass of RCA and RFA per cubic FRAC. The values of *m_g_* and *m_s_* are equal to the product of the volume and apparent density of RCA and RFA per cubic FRAC. From Table 4, the apparent densities of RCA and RFA are 2734.8 kg/m^3^ and 2594.7 kg/m^3^. Putting these data into the Equation (2), *β_s_* = 39.4%. Therefore, the sand ratio in this test is taken as 40%. Then, based on the *w*/*b* ratio and sand-cement ratio obtained from the performance test of mortar, the mix design of FRAC is calculated.

### 4.2. Mixture Design of FRAC

According to the optimum sand-cement ratio determined in the recycled mortar test, the sand ratio was 40%, the water consumption per cube was 200 kg, and the water-reducing agent dosage was 3% of the mass of cement. The test scheme of FRAC with a strength class of 30 MPa was designed with the *w*/*b* ratio and fly ash replacement ratio as the main test variables. The *w*/*b* ratios were 0.3, 0.35, and 0.40, and the fly ash replacement ratios were 0, 20%, and 40% for a total of nine groups of specimens. The slump of these nine groups FRAC were between 50–80 mm. The mixture design and test results of the cube compressive strength, splitting tensile strength, and flexural strength of these nine groups FRAC are shown in Table 7.

### 4.3. Compressive Strength

#### 4.3.1. Effect of *w*/*b* Ratio

The variation of the compressive strength of FRAC with the *w*/*b* ratio is shown in Figure 7. When the *w*/*b* ratio increases, the variation trend of compressive strength is consistent. The compressive strength of FRAC increases first as the *w*/*b* ratio increases from 0.3 to 0.35, and subsequently decreases as the *w*/*b* ratio grows to 0.4. The increase in range was the highest when fly ash was not incorporated. The compressive strength of FRAC reached a climax when the *w*/*b* ratio was 0.35 and without adding fly ash. It is generally believed that the compressive strength of concrete will decrease with the increase of the *w*/*b* ratio. In this paper, the compressive strength of FRAC with a *w*/*b* ratio of 0.35 is higher than the other two types, while the compressive strength of recycled mortar with a water-binder ratio of 0.35 is lower than that of recycled mortar with a water-binder ratio of 0.3. The reason is that, when the *w*/*b* ratio is low, mortar is not sufficient to fully coat all RCA and the bonding effect between mortar and RCA is poor, thus compressive damage is mainly caused by the damage of mortar, the compressive strength of mortar, and is close to FRAC, even slightly higher than FRAC.

#### 4.3.2. Effect of Fly Ash

According to the data in Table 7, the effect of the fly ash replacement ratio on the compressive strength of FRAC is shown in Figure 8. The compressive strength of FRAC decreases with the increase of the fly ash replacement ratio, except when the *w*/*b* ratio is 0.3; as the replacement ratio increases from 0 to 20%, the compressive strength of FRAC hardly changes. However, the compressive strength of FRAC with 40% fly ash decreases by 20.9%. When the *w*/*b* ratio is 0.35, the compressive strength reduces by 7.7% as the fly ash replacement ratio increases to 20%. the compressive strength of FRAC with 40% fly ash further decreases, by comparison with the FRAC without fly ash, and the compressive strength is reduced by 27.2%. When the *w*/*b* ratio increases to 0.4, the compressive strength decreases by 12.3% when the fly ash replacement ratio increases from 0 to 20%. When the replacement ratio increases to 40%, the compressive strength decreases by 24%, compared with FRAC without fly ash. Past studies have indicated that the compressive strength of concrete mixed with fly ash is quite different from that of plain concrete at 28 days, and the difference is gradually decreased at 90 days. The study by Saha et al. [35] indicated that the compressive strength of class F fly ash (calcium content 0.6%) concrete at 28 days decreased, and decreased sharply with the increase of the fly ash replacement ratio. The compressive strength of concrete with 20% and 40% fly ash at 28 days was 68.4% and 52.6% of the compressive strength of plain concrete. As the curing day increases to 90 days, the compressive strength development was 79% and 72.1% of the plain concrete compressive strength for the concrete with 20% and 40% fly ash. For FRAC, Corinaldesi et al. [36] studied the compressive strength of FRAC using 20% fly ash (calcium content 3.08%) and the cement at a 0.40 *w*/*b* ratio is 14.0% higher than plain FRAC. An investigation by Saravanakumar et al. [37] showed that the compressive strengths of FRAC mixed with 40%, 50%, and 60% fly ash (calcium content 1.07%) are 81.1%, 79.2%, and 65.8% of plain FRAC. Kurda et al. [38] calculated and fitted the compressive strength of 665 concrete mixed with fly ash and recycled aggregate. Fitting results showed that, when the replacement rate of coarse and fine recycled aggregate is 100%, the compressive strengths are 112.7% and 76.2% of the plain concrete compressive strength for the concrete with 20% and 40% fly ash. In this paper, only when the *w*/*b* ratio is 0.3, is the compressive strength of FRAC with 20% fly ash higher than that of plain FRAC. The ratio of the compressive strength of 40% fly ash FRAC to the compressive strength of plain FRAC is close to fitting the results of Kurda. The compressive strength ratios of 40% fly ash FRAC to plain FRAC under three *w*/*b* ratios are 79.0%, 72.8%, and 76.0%, respectively.

The effect of fly ash on the mechanical properties of FRAC is influenced to a large extent by its fineness and content of calcium-containing compounds. In general, high-calcium fly ash reacts faster and provides better early-strength, while low-calcium fly ash reacts slower in the early ages of the hydration reaction because of the presence of more chemically inert crystalline phases. The study by Hemalatha et al. [22] has shown that there is still unreacted fly ash and there are incompletely hydrated particles in concrete mixed with low-calcium fly ash after curing 28 days. The fineness of fly ash used in concrete also is a factor which influences the properties of concrete [39]. Chindaprasirt et al. [40] suggested that the packing and nucleation effects in the hydration reaction of fly ash mortar are mainly determined by the fineness of fly ash. Their research also showed that fine fly ash can enhance compressive strength and reduce shrinkage. On the other hand, coarse fly ash is less reactive, requires more water, and creates more pores inside the mortar, causing a decrease in the strength of mortar and concrete [41]. In this experiment, the fly ash used has a lower content (3.36%) of calcium-containing compounds and a coarser fineness, which leads to a decrease in the compressive strength of FRAC. Furthermore, the reduction of compressive strength is relatively low as the *w*/*b* ratio is 0.4, because the water is enough at this time and the reaction of the fly ash is better, so the reduction is smaller.

#### 4.3.3. The Relationship of Compressive Strength between FRAC and Recycled Mortar

The relationship between the compressive strength of FRAC and mortar is shown in Table 8 and Figure 9. It is obvious that the compressive strength of most of the FRAC is higher than that of the recycled mortar, which illustrated that the RCA plays a good role in supporting the compressive damage process. This also shows that FRAC made by this mixture design method has good compressive strength. For recycled mortar, the compressive strength decreases as the *w*/*b* ratio increases, whereas for FRAC, the compressive strength is at a maximum when the *w*/*b* ratio is 0.35. The ratio of the compressive strength of FRAC to the recycled mortar is the largest at this *w*/*b* ratio; this is because the damage of FRAC is caused by the damage of RAC rather than mortar when the *w*/*b* ratio is 0.3.

### 4.4. Splitting Tensile Strength

#### 4.4.1. Effect of *w*/*b* Ratio

From Figure 10, the variation between the splitting tensile strength and the *w*/*b* ratio shows a tendency of first increasing and, succeeding that, decreasing as the *w*/*b* ratio increases. When the *w*/*b* ratio increases from 0.3 to 0.35, the corresponding increasing rates of splitting tensile strength were 21.2%, 19%, and 5.8% at the fly ash replacement ratios of 0, 20%, and 40%, respectively. As the *w*/*b* ratio grows to 0.4, the corresponding decreasing rates of splitting tensile strength were 12.6%, 19.1%, and 17.2% when the fly ash contents were 0, 20%, and 40%, respectively. From Figure 11, the splitting tensile destruction section is relatively smooth, and it is obvious that the RCAs in the yellow circles are split in two along the destruction section, which shows that RCAs bear the main tensile stress in splitting tensile failure.

#### 4.4.2. Effect of Fly Ash

Figure 12 sketches the variation trend of splitting tensile strength with the fly ash replacement ratio. It is manifest that the splitting tensile strength decreases almost linearly when the fly ash replacement ratio increases continuously. When the *w*/*b* ratio is 0.35, the splitting tensile strength value is the highest and the decrease amplitude is the largest, when the fly ash replacement ratios are 20% and 40%, compared with that without fly ash, and the splitting tensile strengths decrease by 6.9% and 21.5%, respectively. The splitting tensile strength decreases the most slowly as the *w*/*b* ratio is 0.3, compared with that without fly ash; the splitting tensile strengths decrease by 5.2% and 10.1% as the fly ash replacement ratios increase to 20% and 40%. As the *w*/*b* ratio increases to 0.4, compared to the splitting tensile strength of FRAC without fly ash, the decreased rates are 13.7% and 25.5% as the fly ash replacement ratios of FRAC are 20% and 40%, respectively. The study by Hashmi et al. [42] indicates that, as the *w*/*b* ratio is 0.45, the splitting tensile strengths of fly ash concrete with 25%, 40%, and 60% of fly ash are observed as 86%, 79%, and 72% of plain concrete at 28 days. In this paper, when the *w*/*b* ratio is 0.4, the splitting tensile strengths of fly ash concrete with 20% and 40% of fly ash are 86.3% and 74.5% of plain concrete, which is close to results of Hashmi. Kurda et al. [38] who collected and fitted tensile strengths of RAC incorporated with fly ash. Fitting results showed that the tensile strength decreased with the increasing incorporation contents of fly ash. For FRAC, the tensile strengths of FRAC with 20% and 40% fly ash are 90.7% and 84.0% of plain concrete. In this paper, when the *w*/*b* ratio is 0.35, the splitting tensile strengths of FRAC with 20% and 40% fly ash are 93.1% and 78.5% of plain FRAC, which is closest to the fitting result. Furthermore, the effect of fly ash on the splitting tensile strength of FRAC is affected by the *w*/*b* ratio. It can be seen that, when the *w*/*b* ratio is relatively low, using fly ash instead of cement has less of an effect on splitting tensile strength compared to FRAC with a high *w*/*b* ratio.

#### 4.4.3. The Relationship of Splitting Tensile Strength and Compressive Strength

The ratio of the splitting tensile strength to compressive strength of FRAC is listed in Table 9. The ratio ranges from 6.4% to 8.7%, which is similar to the result reported by Tangchirapat et al. [43], but lower than the ratio of NAC (approximately 10%). However, the ratio is influenced by the *w*/*b* ratio; with a *w*/*b* ratio of 0.4, the value reaches the maximum. The reason is that, when the *w*/*b* ratio is 0.4, the compressive strength of FRAC is much lower than other groups, while the value of the splitting tensile strength under different *w*/*b* ratios changes little, thus making the value of this ratio change more.

For the conversion formula for splitting tensile strength, the American Concrete Institute (ACI) [44] recommends the following:(3)$$f_{t,sp}=0.59\sqrt{f_{c}^{\prime }}$$
where $f_{t,sp}$ and $f_{c}^{\prime }$ are the splitting tensile strength and cylinder compressive strength, respectively, and the compressive strength of a standard cylinder is 1.05 times that of a cube.

The conversion formula for calculating the splitting tensile strength of ordinary concrete and high-strength concrete given by the Chinese standard is as follows [45]:(4)$$f_{t,sp}=0.19f_{cu}^{0.75}$$

Thus, the conversion formula between splitting tensile strength and compressive strength is assumed to be as following:(5)$$f_{t,sp}=Af_{cu}^{B}$$

The singular point in the data is removed, and the values of *A* and *B* are obtained by fitting. The value of the goodness of fit (R^2^) is 0.839. Therefore, the model is valid. The estimated value of the splitting tensile strength of FRAC can be calculated by the compressive strength by the following conversion formula:(6)$$f_{t,sp}=0.265f_{cu}^{0.663}\;(R^{2}=0.839)$$

The relative differences calculated by the following formula are listed in Table 9. From Table 9, it is illustrated that, when the *w*/*b* ratio is 0.35 and 0.4, and the relative difference is small, the calculated value of the fitting formula is in keeping with the experimental value.
(7)$$\mathrm{Releative}\;\mathrm{difference}=\frac{\mathrm{Calculated}\;\mathrm{value}-\mathrm{Experimental}\;\mathrm{value}}{\mathrm{Experimental}\;\mathrm{value}}$$

### 4.5. Flexural Strength

#### 4.5.1. Effect of *w*/*b* Ratio

The effect of the *w*/*b* ratio on the flexural strength of FRAC is shown in Figure 13. Without the admixture of fly ash, the flexural strength decreases as the *w*/*b* ratio increases. When the *w*/*b* ratio increases from 0.3 to 0.35, the flexural strength decreases slightly, and the decreasing rate is 1.8%. When the *w*/*b* ratio increases from 0.35 to 0.4, the flexural strength decreases obviously by 25%. When the fly ash replacement ratios are 20% and 40%, the flexural strengths increase first and then decrease as the *w*/*b* ratios increase. When the *w*/*b* ratio increases from 0.3 to 0.35, the flexural strengths of FRAC with 20% and 40% fly ash replacement ratios increase by 8.3% and 17.5%, respectively. In the same fly ash replacement ratio, compared to FRAC with a *w*/*b* ratio of 0.3, the flexural strengths of FRAC with a *w*/*b* ratio of 0.4 decrease significantly by 25% and 21.3%, respectively. The flexural strength of most concrete tended to decrease at low *w*/*b* ratios [46]. In this test, the relationship between flexural strength and *w*/*b* ratio is similar to that of compressive strength and also proves that the mechanical properties of FRAC at a *w*/*b* ratio of 0.35 are the most excellent.

#### 4.5.2. Effect of Fly Ash

Figure 14 sketches the effect of the fly ash replacement ratio on the flexural strength of FRAC. According to this figure, with the increase of the fly ash replacement ratio, flexural strength presents a linear downward trend, and the decline amplitude gradually slows down with the increase of the *w*/*b* ratio. When the *w*/*b* ratio is 0.3, the decrease is the most obvious, compared with that without fly ash, the flexural strength of FRAC with 20% and 40% fly ash replacement ratios decrease by 15.8% and 29.8%, respectively. When the *w*/*b* ratio is 0.35 and the fly ash replacement ratios are 20% and 40%, the flexural strengths decrease by 7.1% and 16.1%, respectively. As the *w*/*b* ratio increases to 0.4, the values of the decreasing rates of flexural strength of FRAC with 20% and 40% replacement further reduce to 7.1% and 11.9%, respectively. The use of fly ash exerts a negative influence on the flexural strength of FRAC, and the effect is more pronounced at a low *w*/*b* ratio. The study by Barbuta et al. [47] showed that when the *w*/*b* ratio of is under 0.48, the flexural strength reaches the highest value when the concrete is mixed with 10% fly ash, which is 20.3% higher than that of plain concrete. The flexural strengths of concrete with 20% and 40% fly ash are obtained as 84.1% and 57.1% of plain concrete. An investigation by Siddique et al. [48] showed that the flexural strength decreased by 39%, 48%, and 56% as the cement was replaced with 35%, 45%, and 55% of class F fly ash. In this experiment, although the flexural strength of FRAC decreases with the increase of fly ash, the decrease is far less than the results of the above two tests.

#### 4.5.3. The Relationship of Flexural Strength and Compressive Strength

For the conversion formula between the flexural strength of ordinary concrete and the compressive strength of a standard cylinder, the American Concrete Institute (ACI) [49] recommends taking it as the following:(8)$$f_{tf}=0.62\sqrt{f_{c}^{\prime }}$$

Xiao and Tavakoli et al. [50,51] deem that the above standard is no longer applicable to the conversion formula between the flexural strength and compressive strength of RAC. The flexural strength and compressive strength values are linearly correlated. After a stepwise regression fitting, the flexural strength of FRAC can be expressed as the following:(9)$$f_{tf}=0.25f_{cu}^{0.80}\;(R^{2}=0.832)$$

The R^2^ value of this model is 0.832, which means that the compressive strength can explain 83.2% of the causes of variation in flexural strength. The relative difference between the calculated values obtained from this fitting formula and the experimental values is calculated by Equation (7) and included in Table 10. Most of the relative differences are within 10%, and the fitted formula is able to predict the value of flexural strength.

## 5. Conclusions

A novel mixture design method based on the workability of fully recycled aggregate concrete (FRAC) is proposed and verified by this experiment. The results show that the recycled mortar and FRAC produced by this method have a good strength. On this basis, the effects of the water-binder (*w*/*b*) ratio and class F fly ash on the mechanical properties recycled mortar and fully recycled aggregate concrete (FRAC) were studied. Based on the experimental results of this study, the following conclusions can be drawn:

The mixture design method of FRAC is mainly divided into three steps. Firstly, the optimal sand-cement ratio was determined by the expansion test of recycled mortar. Secondly, the sand ratio was calculated through the “Mortar abundant coefficient method”. Finally, the amount of the FRAC material component can be directly obtained by the “Absolute volume method”. This mixture design method can also be used for the preparation of recycled mortarThe recycled mortar produced by the new method has good fluidity and strength, and the strength can reach 85.7% to 92% of the standard mortar strength. In addition, the compressive strength ratio of FRAC to recycled mortar is mostly greater than 1, which shows that it is feasible to fabricate FRAC by this method.The addition of class F fly ash can ameliorate the fluidity of recycled mortar, but it will reduce compressive strength; especially for mortar with a larger water-binder ratio, the effect to improve the fluidity is reduced, and the reduction of compressive strength is increased.The mechanical properties of FRAC are best, as the water-binder ratio is 0.35. The ratio of compressive strength between recycled mortar and FRAC is from 0.84 to 1.22 for the same ratio, while the splitting tensile strength of FRAC is 6.4% to 8.7% of the compressive strength for the same ratio, which is less than that of NAC.The addition of class F fly ash reduces the mechanical properties of FRAC, and the reduction increases almost linearly with the replacement ratio of fly ash. The main reason is that the calcium oxide content of fly ash is low, and the particle size is large; fly ash cannot be fully reacted at 28 days, resulting in a decrease in strength. Therefore, the strength evaluation time of FRAC with fly ash should be 90 days or longer. The effect of class F fly ash on the compressive strength and splitting tensile strength of FRAC is close to the previous experimental results, while the effect on the flexural strength of FRAC is lower than the previous experimental results. The large-scale application of class F fly ash in FRAC requires the further treatment of fly ash or the addition of other auxiliary materials.The theory conversion formulas between the splitting tensile strength, flexural strength, and compressive strength of FRAC are deduced, respectively, and the results indicate that the compressive strength can explain more than 80% of the causes of variation in splitting tensile strength and flexural strength.

## Figures and Tables

**Figure 1 materials-15-08143-f001:**
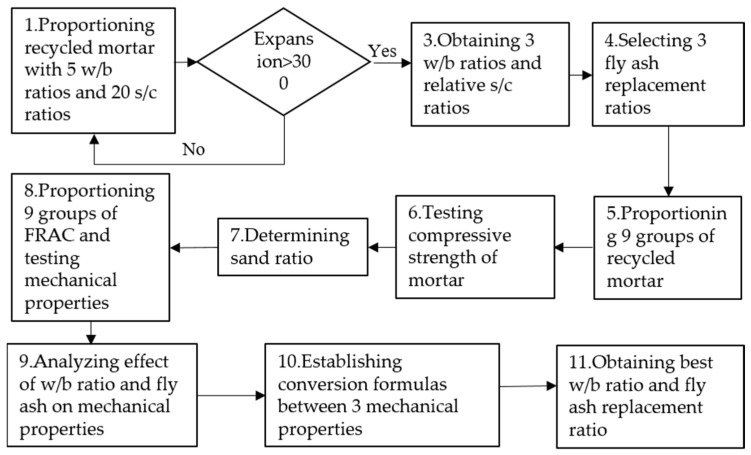
Research steps.

**Figure 2 materials-15-08143-f002:**
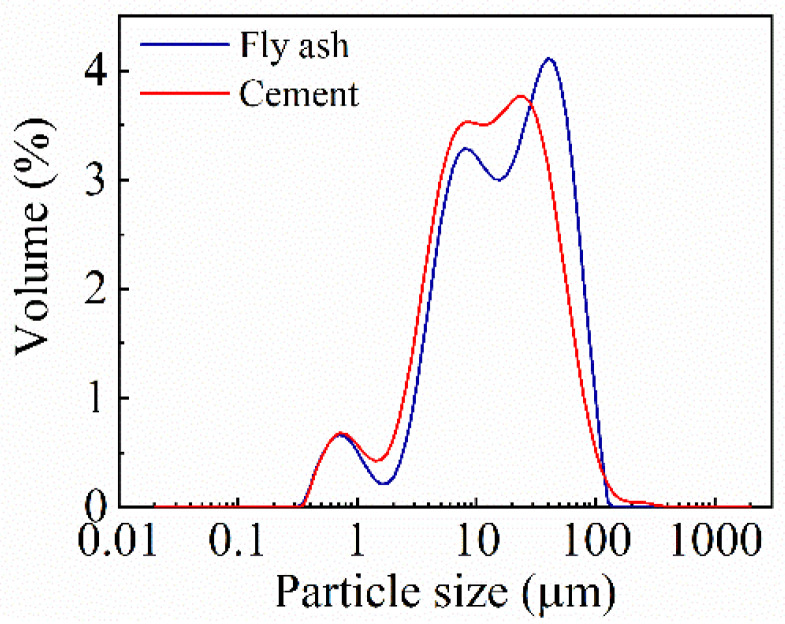
Particle size distribution of binder materials.

**Figure 3 materials-15-08143-f003:**
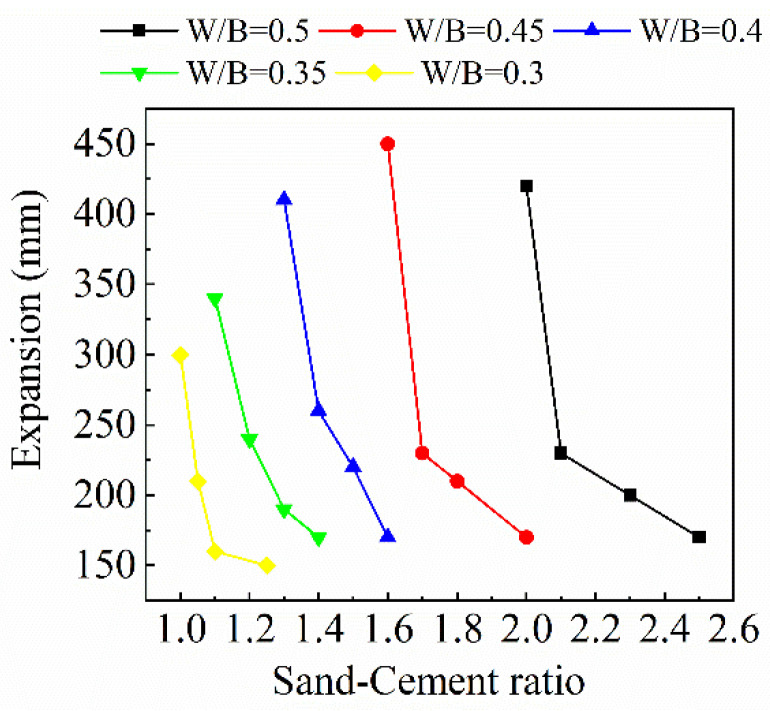
Variation of expansion with sand-cement ratio under various *w*/*b* ratio.

**Figure 4 materials-15-08143-f004:**
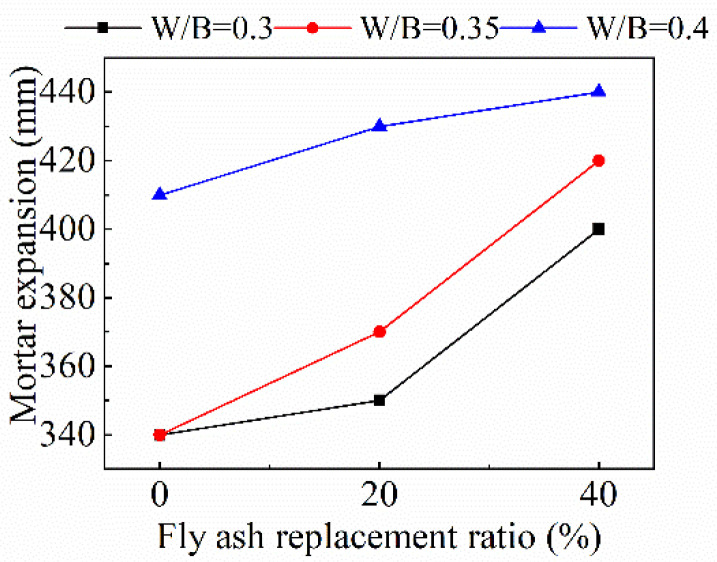
Variation of expansion with fly ash replacement ratio.

**Figure 5 materials-15-08143-f005:**
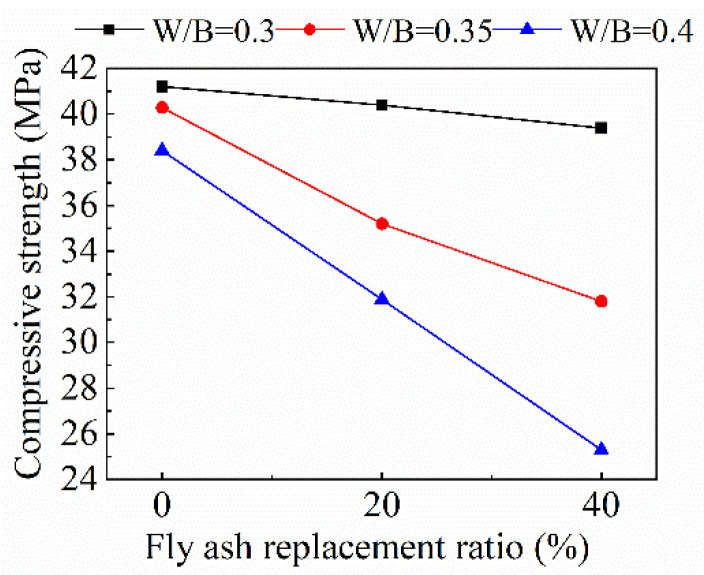
Variation of recycled mortar compressive strength with fly ash replacement ratio.

**Figure 6 materials-15-08143-f006:**
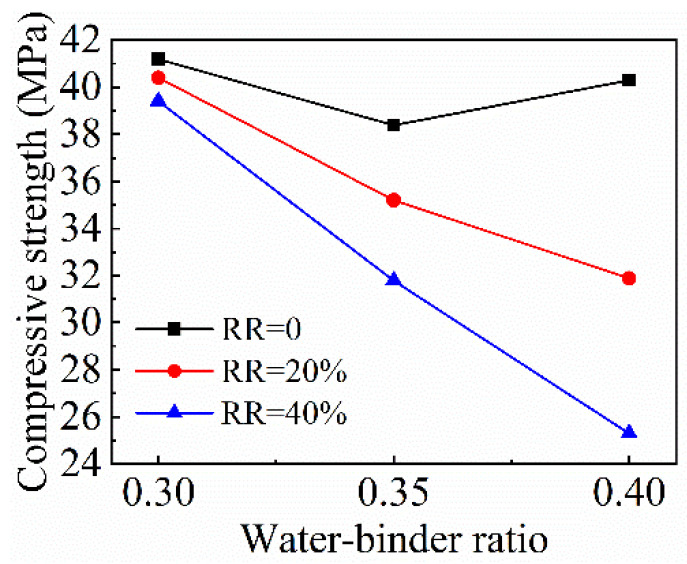
Variation of recycled mortar compressive strength with *w*/*b* ratio. RR represents replacement ratio of fly ash.

**Figure 7 materials-15-08143-f007:**
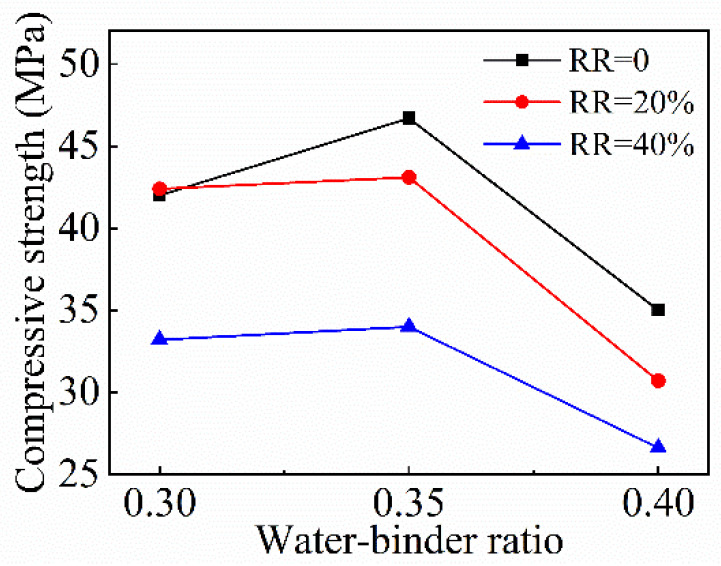
Variation of FRAC compressive strength with *w*/*b* ratio.

**Figure 8 materials-15-08143-f008:**
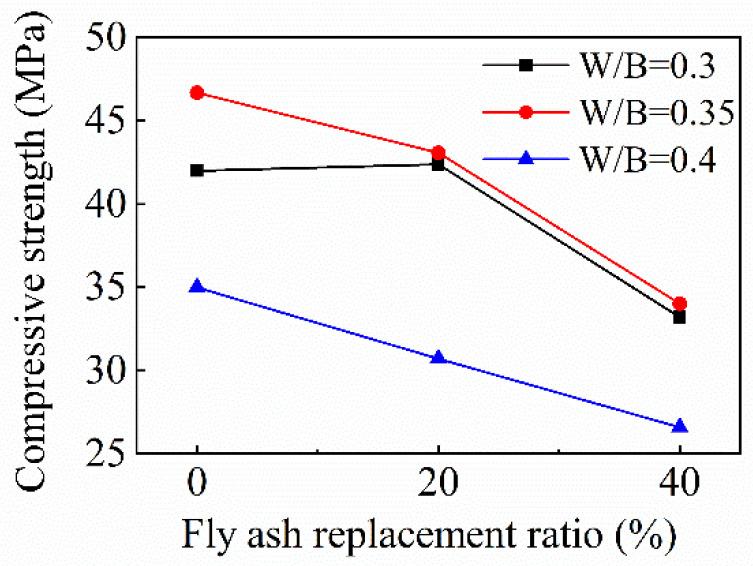
Variation of FRAC compressive strength with fly ash replacement ratio.

**Figure 9 materials-15-08143-f009:**
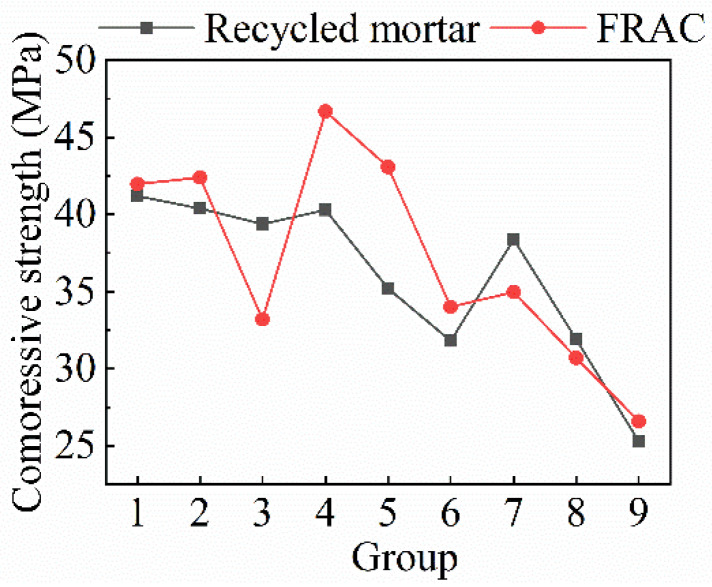
The relationship between mortar strength and compressive strength of FRAC.

**Figure 10 materials-15-08143-f010:**
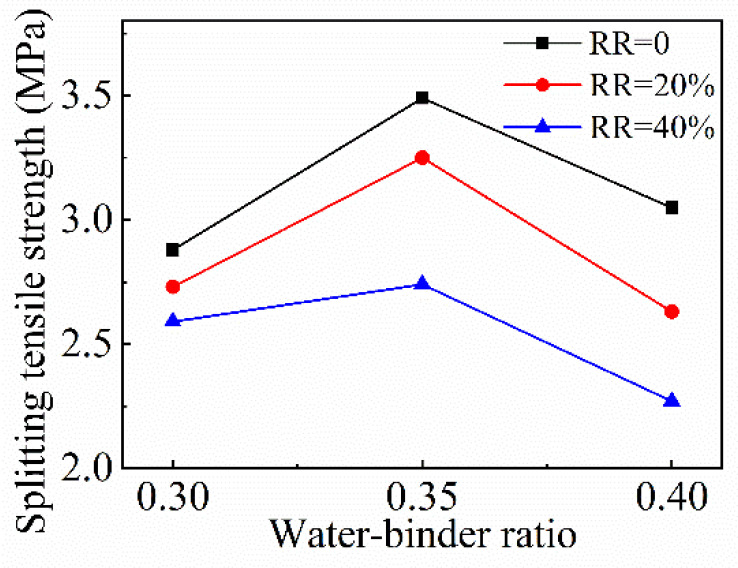
Variation of FRAC splitting tensile strength with *w*/*b* ratio.

**Figure 11 materials-15-08143-f011:**
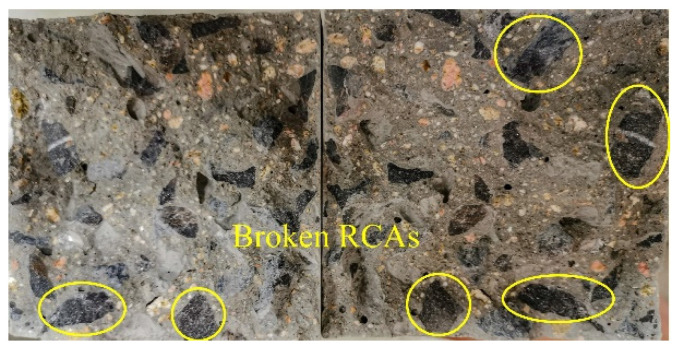
The typical destruction section of FRAC after splitting tensile destruction.

**Figure 12 materials-15-08143-f012:**
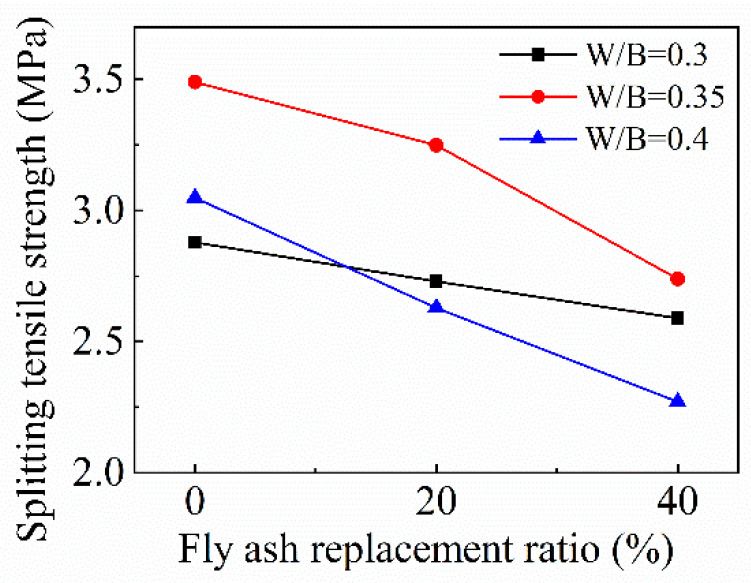
Variation of FRAC splitting tensile strength with fly ash replacement ratio.

**Figure 13 materials-15-08143-f013:**
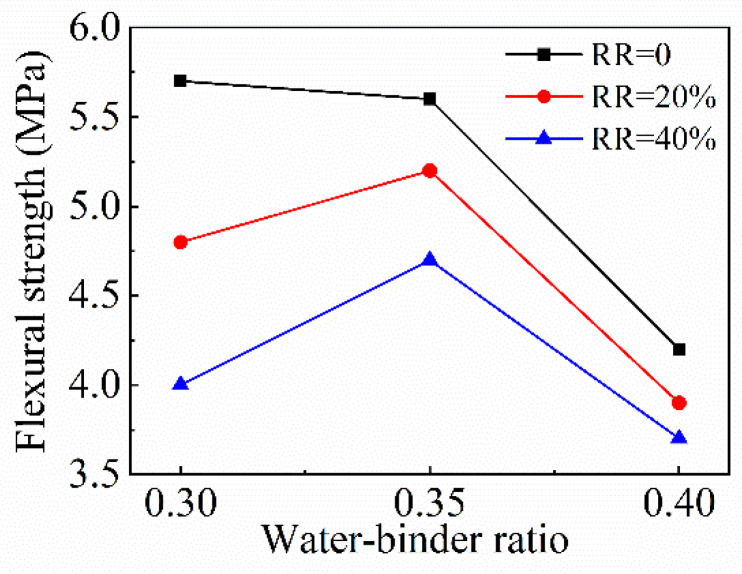
Variation of FRAC flexural strength with *w*/*b* ratio.

**Figure 14 materials-15-08143-f014:**
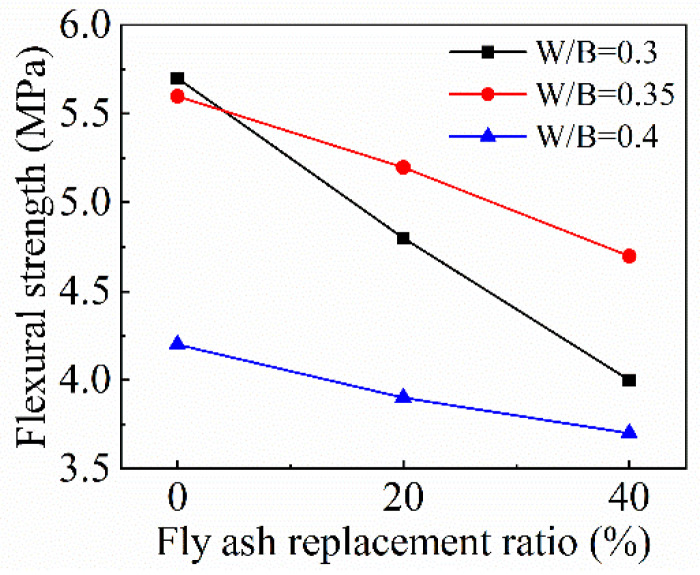
Variation of FRAC flexural strength with fly ash replacement ratio.

**Table 1 materials-15-08143-t001:** The physical properties of binder materials.

	Density (kg/m^3^)	Packing Density (kg/m^3^)	Specific Surface Area (m^2^/kg)	Water Consumption at Standard Consistency (%)	28 d Compressive Strength (MPa)
Cement	3126	1550	356	27	44.8
Fly ash	2550	1120			

**Table 2 materials-15-08143-t002:** Main chemical composition of binder materials.

Chemical Composition (%)	SiO_2_	CaO	Al_2_O_3_	Fe_2_O_3_	MgO	SO_3_	K_2_O	TiO_2_
Cement	27.73	46.31	13.54	3.09	3.09	2.82	0.984	0.688
Fly ash	54.74	3.36	33.33	2.30	0.867	0.933	2.21	1.01

**Table 3 materials-15-08143-t003:** Particle size distribution of cement and fly ash.

Particle Size (μm)	Surface Weighted Mean D	Vol. Weighted Mean	d (0.1)	d (0.5)	d (0.9)
Cement	5.194	19.999	2.691	12.673	46.709
Fly ash	5.692	24.373	3.307	16.502	57.635

Note: d (0.1), d (0.5), and d (0.9) represent the diameters corresponding to 10%, 50%, and 90% of the cumulative size distribution (0 to 100%), respectively.

**Table 4 materials-15-08143-t004:** Properties of RCA and RFA.

Aggregate	Apparent Density (kg/m^3^)	Water Absorption (%)	Crushing Index (%)	Fineness Modulus	Loose Stack Density (kg/m^3^)	Compact Stack Density (kg/m^3^)	Void Ratio (%)
RCA	2734.8	3.56	14.1	-	1291.8	1378.7	52.8
RFA	2594.7	7.6	-	3.15	1486		

**Table 5 materials-15-08143-t005:** The relationship between expansion and sand-cement ratio under different w/b ratio.

Mortars NO	*W*/*B*	S/C	Expansion (mm)
M-0.5	0.5	2.5	170
		2.3	200
		2.1	230
		2	420
M-0.45	0.45	2	170
		1.8	210
		1.7	230
		1.6	450
M-0.4	0.4	1.6	170
		1.5	220
		1.4	260
		1.3	430
M-0.35	0.35	1.4	170
		1.3	190
		1.2	240
		1.1	340
M-0.3	0.3	1.25	150
		1.1	160
		1.05	210
		1	300

**Table 6 materials-15-08143-t006:** Mixture design and compressive strength of recycled mortar.

Mortars NO	Cement (g)	Fly Ash (g)	Water (g)	Water-Reducing Agent (g)	Sand (g)	S/C	Expansion (mm)	*f_cu_* (MPa)
M0.30F0	1100	0	330	33	1100	1	300	41.2
M0.30F20	880	220	330	27	1100	1	340	40.4
M0.30F40	660	440	330	22	1100	1	400	39.4
M0.35F0	1100	0	385	33	1210	1.1	340	38.4
M0.35F20	880	220	385	27	1210	1.1	370	35.2
M0.35F40	660	440	385	22	1210	1.1	420	31.8
M0.40F0	1100	0	440	33	1430	1.3	410	40.3
M0.40F20	880	220	440	27	1430	1.3	430	31.9
M0.40F40	660	440	440	22	1430	1.3	440	25.3

**Table 7 materials-15-08143-t007:** Mixture design (kg/m^3^) and strength (MPa) of recycled concrete.

NO	Specimen ID	Water	Cement	Fly Ash	S/C	RFA	RCA	WRA	*f_cu_*	*f_t,sp_*	*f_tf_*
1	C0.30F0	200	600	0	1	600	900	18	42.0	2.88	5.7
2	C0.30F20	200	480	120	1	600	900	14.4	42.4	2.73	4.8
3	C0.30F40	200	360	240	1	600	900	10.8	33.2	2.59	4.0
4	C0.35F0	200	571	0	1.1	628	942	17.1	46.7	3.49	5.6
5	C0.35F20	200	457	114	1.1	628	942	13.7	43.1	3.25	5.2
6	C0.35F40	200	343	228	1.1	628	942	10.3	34.0	2.74	4.7
7	C0.40F0	200	500	0	1.3	650	975	15	35	3.05	4.2
8	C0.40F20	200	400	100	1.3	650	975	12	30.7	2.63	3.9
9	C0.40F40	200	300	200	1.3	650	975	9	26.6	2.27	3.7

Note: S/C represents sand-cement ratio, WRA represents water-reducing agent, *f_cu_* represents cube compressive strength, *f_t,sp_* represents splitting tensile strength, *f_tf_* represents flexural strength.

**Table 8 materials-15-08143-t008:** Ratio of compressive strength (MPa) between FRAC and mortar.

NO	Specimen ID	Compressive Strength of FRAC	Compressive Strength of Mortar	Ratio
1	C0.30F0	42.0	41.2	1.02
2	C0.30F20	42.4	40.4	1.05
3	C0.30F40	33.2	39.4	0.84
4	C0.35F0	46.7	40.3	1.16
5	C0.35F20	43.1	35.2	1.22
6	C0.35F40	34.0	31.8	1.07
7	C0.40F0	35	38.4	0.91
8	C0.40F20	30.7	31.9	0.96
9	C0.40F40	26.6	25.3	1.05

**Table 9 materials-15-08143-t009:** Ratio of splitting tensile strength to compressive strength and relative difference.

NO	Specimen ID	*f_cu_*	Experimental Values of *f_t,sp_*	*f_t,sp/_f_cu_*	Calculated Values of *f_t,sp_*	Relative Difference
1	C0.30F0	42.0	2.88	6.9%	3.16	10%
2	C0.30F20	42.4	2.73	6.4%	3.18	16%
3	C0.30F40	33.2	2.59	7.8%	2.70	4%
4	C0.35F0	46.7	3.49	7.5%	3.39	−3%
5	C0.35F20	43.1	3.25	7.5%	3.21	1%
6	C0.35F40	34.0	2.74	8.1%	2.75	0
7	C0.40F0	35	3.05	8.7%	2.80	−8%
8	C0.40F20	30.7	2.63	8.6%	2.57	−2%
9	C0.40F40	26.6	2.27	8.5%	2.33	3%

**Table 10 materials-15-08143-t010:** Relative difference of calculated value and experimental value of flexural strength.

NO	Specimen ID	*f_cu_*	Experimental Values of *f_tf_*	Calculated Values of *f_tf_*	Relative Difference
1	C0.30F0	42.0	5.7	4.97	−13%
2	C0.30F20	42.4	4.8	5.01	4%
3	C0.30F40	33.2	4.0	4.12	3%
4	C0.35F0	46.7	5.6	5.41	−3%
5	C0.35F20	43.1	5.2	5.08	−2%
6	C0.35F40	34.0	4.7	4.20	−11%
7	C0.40F0	35	4.2	4.30	2%
8	C0.40F20	30.7	3.9	3.87	−1%
9	C0.40F40	26.6	3.7	3.45	−7%

## Data Availability

All experimental data in this paper have been displayed in the Table of the paper.

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
