# Peer review of "Mixture Design and Mechanical Properties of Recycled Mortar and Fully Recycled Aggregate Concrete Incorporated with Fly Ash"

_materials, 2022, doi:10.3390/ma15228143_

Round 1

Reviewer 1 Report

The authors have present a study to address the procedure for mix design of fully recycled aggregate concrete and discussed the mechanical properties of resulting concrete.  However, the details related to the mix design are very limited. The main novelty of the current study appears to be the mix design procedure which is not fully described. The authors should present a flow chart, clearly stating the steps involved in the procedure. In addition, few test like expansion test have not been defined in the methodology.  A lot of studies have already been done on the recycled concrete and mortar and the other parameters studied in this study (the influence of w/b ratio and fly ash replacement ratio on mechanical properties) have also been examined a lot, therefore, the main novelty for this study should be targeted and discussed. In addition the introduction section needs lot of improvement; please add more reference to support the research statement. The studies from literature should also be added in the results section to justify the results. Proof reading by native English speaker for rephrasing of various sentences is also required for the manuscript. 

The above comments should be considered before the final acceptance of this paper.

Author Response

Thank you for your affirmation to our paper. We have reread the manuscript and revised and rewritten it. Tests of mortar expansion have been defined, corresponding to Line 139-144 in new manuscript.

Introduction and statement have been rewritten, and more reference have been added. We added Figure.1 to explain the research methods.

In the discussion of the results, the previous research results are added and compared with the results of this paper.

Finally, the effect of fly ash in the conclusion is rewritten and the grammatical errors in the manuscript are corrected.

Reviewer 2 Report

The manuscript studies the effect of the water/binder ratio in fully recycled concretes incorporating fly ash and waste concrete. Overall, the manuscript does not represent a scientific breakthrough in this field of research, there are many previous researches that have already analysed this. On the other hand, the article is well written and justified and presents results that can partly contribute to the development of this type of new concrete.

As a general comment, some reference to the use of fly ash should be included in the title as this concept is of vital importance in the whole development of the research.

On the other hand, some more specific comments are made:

Line 92. Define P.O.

Line 96. Define whether fly ash substitution was by weight or by volume. Fly ash has a different density than cement

Line 101 Clarify what is meant in this paragraph "The water-reducing rate of polycarboxylic acid water-reducing agent used in the experiment is 27%".

Table 3. Define d (0.1), d (0.5) and d (0.9).

Table 4. Indicate the standards that have been followed for the calculation of the physical properties of RA.

Line 114. The authors can give more details on how the expansion test was carried out, was it according to the standard?

Line 224. Where did you obtain these values? Give some reference

Line 283. 28 days space is missing.

Line 290. According to the authors in this line, fly ash has a higher fineness than cement. But from Figure 1 and Table 3 it appears that fly ash has a higher particle size than cement. Why do they make this statement?

Figure 10. Improve quality. What is described in the text is not apparent.

Lines 269 -275, Lines 331-335 and Lines 396-403. Can you provide some previous references to the drops, are the results observed in your research in line with those obtained by other authors?

Author Response

Part B (Reviewer 2).

Main Comment

The reviewer’s comment: The manuscript studies the effect of the water/binder ratio in fully recycled concretes incorporating fly ash and waste concrete. Overall, the manuscript does not represent a scientific breakthrough in this field of research, there are many previous researches that have already analysed this. On the other hand, the article is well written and justified and presents results that can partly contribute to the development of this type of new concrete.

As a general comment, some reference to the use of fly ash should be included in the title as this concept is of vital importance in the whole development of the research.

The authors’ answer: Thank you for your affirmation to our paper. Your suggestion is interesting and useful, so we have added some reference to review the effect of fly ash on the properties of recycled concrete (From line 71-85). And we also added discussion and conclusion on the effect of fly ash on recycled concrete and recycled mortar.

Replies to specific comments in this manuscript were presented as following:

1.     Line 92. Define P.O.The authors’ answer: P.O. refers to ordinary Portland cement. Original text has been changed to ‘Grade 42.5 ordinary Portland cement’. 2.      Define whether fly ash substitution was by weight or by volume. Fly ash has a     different density than cementThe authors’ answer: The fly ash replacement ratio is the ratio of the weight of fly ash to the weight of total cementitious material. This has been noted in the manuscript, corresponding to Line 115-116. 3.     Line 101 Clarify what is meant in this paragraph "The water-reducing rate of polycarboxylic acid water-reducing agent used in the experiment is 27%".The authors’ answer: Water-reducing rate refers to the ratio of the difference between the unit water consumption of concrete without water-reducing agent and concrete with water-reducing agent to the unit water consumption of concrete without water-reducing agent when the slump of concrete is basically the same. Its calculation formula is as follows:Where, WR is water-reducing rate, W0 and W1 are the unit water consumptions of concrete without water-reducing agent and concrete with water-reducing agent, respectively. 4.     Table 3. Define d (0.1), d (0.5) and d (0.9).The authors’ answer: d (0.1), d (0.5) and d (0.9) represent the diameters corresponding to 10 %, 50 %, and 90 % of the cumulative size distribution (0 to 100 %), respectively. This has been added in manuscript, corresponding to Line 127-128. 5.     Table 4. Indicate the standards that have been followed for the calculation of the physical properties of RA.The authors’ answer: The physical properties of RA are tested according to Chinese standard JGJ 52-2006. This has been added in manuscript as reference [22], corresponding to Line 119. 6.     Line 114. The authors can give more details on how the expansion test was carried out, was it according to the standard?The authors’ answer: The expansion test was carried out by Chinese standard GB50119-2013, reference [23]. The detailed test procedure has been written into the manuscript, corresponding to Line 139-145. 7.     Line 224. Where did you obtain these values? Give some referenceThe authors’ answer: The letters in Equations 1 and 2 are explained more detail form Line 250 to Line 257. Vg, Vs are the volumes of coarse aggregate and fine aggregate per cubic concrete, respectively. Pg is the void ratio of the RCA, these parameters are all about aggregate performance, and the test methods are shown in Section 2.1. γ is mortar rich coefficient, we added relevant test specifications as reference [28].   8.      Line 283. 28 days space is missing.The authors’ answer: The problem has been modified.  9.     Line 290. According to the authors in this line, fly ash has a higher fineness than cement. But from Figure 1 and Table 3 it appears that fly ash has a higher particle size than cement. Why do they make this statement?The authors’ answer: Thank you for your suggestion. Our expression here is ambiguous. The conclusion in Reference [22] shows that the incorporation of fly ash with large particle size (coarse fineness) will reduce the compressive strength of mortar. ‘higher’ has been changed by ‘coarser’ in the original manuscript to show the larger particle size, corresponding to Line 344. 10.  Figure 10. Improve quality. What is described in the text is not apparent. The authors’ answer: This Figure has been replaced with a clearer one. 11.  Lines 269 -275, Lines 331-335 and Lines 396-403. Can you provide some previous references to the drops, are the results observed in your research in line with those obtained by other authors? The authors’ answer: Comparison between this study and previous studies on fly ash concrete has been added to the manuscript. The changes in compressive strength and splitting tensile strength obtained in this paper are in good agreement with previous studies, while the effect of fly ash on flexural strength is lower than some previous studies.

Round 2

Reviewer 1 Report

Thank you for the authors for addressing the comments, however, the reviewer is of the opinion that the significant changes have not yet have made. The novelty of the paper as mentioned before is the mix design, which should be discussed in detail and the focal point of the discussion should be that area in the light of the chemical composition of the concrete components specially the recycled aggregate which will significantly affect the fresh and hardened properties. The authors have focused on the addition of fly ash including the influence of w/b ratio and fly ash replacement ratio on mechanical properties which is already well established. The main discussion on how the mix design will be affected by different parameters is lacking. The main novelty of the current study appears to be the mix design procedure which is not fully described. The results should explained with regard to mix design otherwise the rest of the title “mechanical properties of recycled mortar and fully recycled aggregate concrete incorporated with fly ash “ is not new, a lot of research has already been done.

The above comments should be considered before the final acceptance of this paper.

Author Response

Thank you for your pertinent suggestion. We have modified the paper according to these advice carefully. The emphasis of this revision is to explain the mix design method and its novelty.

As for the chemical composition of recycled aggregate, you are very to the point. Which has great influence on the mixture design of recycled aggregate. We have done some relevant experiments before and published some papers. These previous research results are mainly reflected in references 14 and 15. We have also added relevant discussions in the introduction.

Section 2.1 “Mixture design method “was added to explain the mixture design method in detail. We hope that the thoughts and methods of mixture design method of FRAC and recycled mortar are more clearly to reader through this statement. The workability of FRAC can be significantly improved by optimal sand-cement ratio and sand ratio, which is the biggest difference from the traditional volume method. We also emphasize that this mix design method are suitable recycled mortar.

The influence of main parameter of mixture design on FRAC properties is reflected in the discussion. Two paragraph about the mix design method is also added to the conclusion.

Reviewer 2 Report

Thank you for making the suggested changes. The quality of the paper has improved considerably.

Author Response

Thank you for your suggestion and affirmation.

Round 3

Reviewer 1 Report

Thank you for addressing the comments.